# The Usage of an Air Purifier Device with HEPA 14 Filter during Dental Procedures in COVID-19 Pandemic: A Randomized Clinical Trial

**DOI:** 10.3390/ijerph19095139

**Published:** 2022-04-23

**Authors:** Paolo Capparè, Raffaele D’Ambrosio, Renato De Cunto, Atanaz Darvizeh, Matteo Nagni, Enrico Gherlone

**Affiliations:** 1Department of Dentistry, IRCCS San Raffaele Hospital, 20132 Milan, Italy; dambrosioraffaele28@gmail.com (R.D.); renatodecu@gmail.com (R.D.C.); atanazdarvizeh@gmail.com (A.D.); nagnimatteo@hotmail.it (M.N.); gherlone.enrico@hsr.it (E.G.); 2Dental School, Vita-Salute San Raffaele University, 20132 Milan, Italy

**Keywords:** COVID-19, HEPA filter, decontamination, aerosols, droplets

## Abstract

The aim of the present study was to evaluate the efficacy of an air purifier device (professional XXl inn-561 innoliving) with HEPA 14 filter in reducing the number of suspended particles generated during dental procedures as a vector of COVID-19 transmission. The survey was conducted on 80 individuals who underwent Oral Surgery with dental Hygiene Procedures, divided into two groups based on the operational risk classification related to dental procedures: a Test Group (with application of filtering device) and a Control Group (without filtering device). All procedures were monitored throughout the clinical controls, utilising professional tools such as molecular particle counters (Lasair III 350 L 9.50 L/min), bacteriological plates (Tryptic Soy Agar), sound meters for LAFp sound pressure level (SPL) and LCpk instantaneous peak level. The rate of suspended particles, microbiological pollution and noise pollution were calculated. SPSS software was used for statistical analysis method. The results showed the higher efficacy of the TEST Group on pollution abatement, 83% more than the Control fgroup. Additionally, the contamination was reduced by 69–80%. Noise pollution was not noticeable compared to the sounds already present in the clinical environment. The addition of PAC equipment to the already existing safety measures was found to be significantly effective in further microbiological risk reduction.

## 1. Introduction

Following the COVID-19 pandemic emergency and its worldwide spread in such a short time, the various anti-contagion strategies and their effectiveness were evaluated in different studies [1]. The person-to-person transmission of SARS-CoV-2 virus like many other microorganisms can be through aerosols; therefore, basic personal protective equipment (PPE) may not be sufficient to prevent it [1].

In addition to basic personal protective equipment such as surgical masks, particle filter respirators (P2 or N95), gloves, goggles, glasses, face shields, gowns and aprons, ventilation control procedures (CDC; Centers for Disease Control and Prevention of United States) are also recommended [2,3]. According to World Health Organization (WHO) regarding COVID-19 transmission risk factors, the virus spreads mostly among people who are in close contact with each other due to short-range aerosol transmission. Therefore, using protective masks (e.g., surgical masks, FFP2, FFP3) plays an important role [4,5]. Thus, at this point, all the possible risk factors present during dental practices most be re-evaluated.

The US CDC guidance has recommended an improvement of ventilation systems and/or adding a portable air cleaner (PAC) in high risk environments such as dental clinics in order to minimize the potential risks associated with aerosols [3]. It is important to notice that during all dental practice activities there is a lack of one or more patient’s PPEs, which creates a risky situation due to the unprotected close contact. One of the main tools in dental practice is air and water jets, which itself produces a noticeable amount of aerosols. Considering the fact that aerosol is the main transmission vector for this virus [4,5], it is clear that the presence of a COVID-positive patients in a closed environment can potentially increase the transmission power as the virus remains in suspension in a greater concentration and for a longer duration.

As the pandemic slows down by the help of anti-COVID vaccination, the places hosting a high turnover of people in a short time still remain under the high risk environments category. This study stems from the need to review the definition of close contact in the dental field and the operational strategies that can be implemented in order to reduce the transmission risk. The optimal solution is to significantly reduce the overall number of suspended particles inside the operating room by utilising an Airpurifier which consists of an HEPA 14 filter, capable of purifying the air in a larger scale than a normal airpurifier. The term HEPA stands for high-efficiency particulate air, indicates a particular highly efficient filtration system for fluids, liquids or gases.

The HEPA filter is made of microfibre filter sheets, thousands of glass fibers that intertwine in multiple layers, separated by aluminum septa. These layers of filter sheets have the task of blocking the polluting particles present in the area to be treated. Many studies have shown that PAC is an effective tool in removing the ultrafine aerosol particles from indoor environments and improving the air quality [6,7,8] in comparison with using solely ventilators [9]. A total number of 1970 studies have shown that using a filter-blower device with a HEPA filter could reduce the microbial concentration peak in dental clinic rooms by 72–97% after 20 min [10]. A recent study used engineering simulations of fluid dynamics by placing a PAC near the patient’s head. This has demonstrated a reduction in aerosol particles present in the respiratory system of first and second dental operators [11].

The aim of this study is to evaluate the effectiveness of PAC devices with HEPA 14 filters as a microbiological abatement system, in order to identify a protocol which guarantees greater safety for both dentist and patient.

## 2. Materials and Methods

The experiment was conducted at the Department of Dentistry, IRCCS San Raffaele Hospital, Milan, Italy. A specific informed consent related to each treatment was obtained from the patients prior to perform any dental procedures.

Based on the operational risk classification related to dental procedures (Gherlone et al., 2021), a total of 80 cases were divided into two groups: 40 in high risk (professional dental hygiene) and 40 in medium risk groups (simple oral surgery procedure; without usage of instruments that generate aerosol e.g., dental turbine). Then, each class was further divided into test and control groups (Figure 1): High Risk Test, High Risk Control, Medium Risk Test and Medium Risk Control [12]. The difference between the Test and Control group was that for the Test group, PAC devices with Hepa 14 filters were on, but for the control group, the PAC devices with the Hepa 14 filter were off.

All cases, in both test and control groups, underwent the same operating conditions (Figure 2).

The study consisted of in vivo operating procedures with and without activation of an air purifier (PAC) with HEPA 14 filter (professional XXL inn-561). All procedures were performed in a dental treatment room at the Dental Clinic of San Raffaele Hospital with 20 m^2^ area and 60 m^3^ volume. The pre-existing ventilation flow for sufficient air change was calculated per hour. Before proceeding to the operational phase, an AT-REST test was carried out without the presence of personnel:-**PHASE 1**: Empty and closed room (sampling for 60 min continuously)-**PHASE 2**: Empty and closed room with PAC on (sampling for 60 min continuously) and then during operational activities with presence of personnel)

During operational activities, the number of particles were calculated using a particle counter system (Lasair III with a flow rate of 50 L per minute, model 350 L), which further discriminates the suspended particles by diameter (0.3; 0.5; 1.0; 5.0 microns). The dBs produced by the device were calculated and microbial quantification was performed during operational phase. Before each procedure (between one patient and another) the same air quality conditions were created by standard sanitization of the environment and the particle concentration was monitored with respect to the baseline value (t0). The PAC device was placed at a distance of 50 cm from the dental chair base and all measurements were taken at the same point (Figure 3 and Figure 4).

The data recorded during the trial were analysed and compared with each other to discriminate between any differences between test and control groups. All tests were performed in an area with the highest pollution degree in both Operational and At-Rest phases with the PAC on and off.

-
**Airborne Aerosol Particles Abatement Power**


The PAC was set to its maximum suction speed (level 4) and was placed behind the patient’s chair (head side) with the intake nozzle facing toward the patient’s head.

-
**Microbiological Eradication Power**


Since the dental clinic environment was presumed to be a high biocontamination area (class C or D of Annex 1 GMP), thus microbiological abatement power has been assessed with the same measurement method for particle abatement. The tests were carried out under operational conditions for both high and medium risk groups (1 and 2), with the PAC system on and off. The final results were reported in Colony Forming Units (CFU) per 1000 L. The aspirated volume was calculated for each run of 250 L per point. Any sort of operator-induced contamination was prevented throughout device mounting and sampling. After sampling, the plates were collected, marked and packaged immediately in clean, refrigerated containers suitable for the transport of biological material; on the day of sampling (and in any case no later than 24 h) the plates were delivered to the test laboratory, where they were incubated for three days to detect aerobic mesophilic bacteria and five days for mold and yeast contamination. The study was aimed at determining the presence of Aerobic Mesophilic Bacteria, Mold and Yeasts using Petri dishes containing Tryptic Soy Agar.

-
**Noise Pollution Degree**


In order to fully characterize the noise degree during the procedures, three parameters were evaluated:(A)LAFp sound pressure level (SPL), with A-frequency weighting profile and F-time constant (Fast). The sound level corrected with the A-weighting is intended to simulate the auditory response for low sound levels; the time constant F (Fast) allows us to measure the instantaneous sound level influenced by the pressure trend in one-eighth of a second (F = 0.125 s). This parameter is the one closest to the auditory sensation detected by the human ear.(B)LCpk instantaneous peak level, with C-frequency weighting profile. This parameter allows us to evaluate the peak level produced by sound pressure in a very short time interval (100 μs); the correct sound level with C-weighting simulates auditory sensation for very high sound levels.(C)A-weighted sound exposure level LAE. This parameter allows you to measure the energy content of sound pressure; in particular it corresponds to the constant sound pressure level with a duration of 1 second, which contains the same energy as the sound event under consideration.

Noise measurements are carried out at the same points where At-Rest environmental contamination measurements were carried out (cf. 10.2). The lit PAC is set at the maximum suction speed (level 4) and was placed behind the patient’s chair (head side) with the intake vent facing the patient’s head.

## 3. Results

### 3.1. Particle Abatement Results

#### 3.1.1. Particles Abatement “At Rest”

In order to assess the air contamination, 6 air samples were collected and monitored from different points of the environment “At Rest” in accordance with UNI EN ISO 14644-1. The results showed the reduction in contamination in two different situations (PAC off and PAC on):-85% ± 1.56% reduction (range: 83–86%) of 0.3 μm particle size-84% ± 1.67% reduction (range: 82–85%) of 0.5 μm particle size-76% ± 4.54% reduction (range: 71–80%) of 1.0 μm particle size-64% ± 17.69% reduction (range: 42–76%) of 5.0 μm particle size

Subsequently, in the two different situations (PAC on and off), a 10 min monitoring was carried out at two critical points (more difficult to clean based on the microbiological contamination results during At REST). The following abatement values were found: -82% ± 1.54% reduction (range: 81–84%) of 0.3 μm particle size-82% ± 1.31% reduction (range: 81–83%) of 0.5 μm particle size-78% ± 8.58% reduction (range: 69–86%) of 1.0 μm particle size-78% ± 21.90% reduction (range: 48–90%) of 5.0 μm particle size

On the basis of the data shown above, it can be noted that for at least three particle sizes (0.3 μm, 0.5 μm and 1.0 μm) the reduction rates obtained during the room sampling at 6 points overlap with those obtained during the 10 min monitoring (Figure 5).

#### 3.1.2. Particles Abatement during “Operational” Phase

To assess the air contamination during the “Operational” phase, monitoring was carried out with PAC on and off, thorough two activities with different risk levels:-Risk activity 1: “Professional Dental Hygiene” identified with letter H-Risk activity 2: “Oral Surgery” procedures identified with letter S

For activity H, the mean value of 3 runs showed a reduction of particle pollution when the PAC device was on (Table 1 and Figure 6):-73% ± 10.57% reduction of 0.3 μm particle size-75% ± 14.28% reduction of 0.5 μm particle size-65% ± 16.37% reduction of 1.0 μm particle size-49% ± 18.78% reduction of 5.0 μm particle size

The findings demonstrated that a reduction in aerosol particle pollution can be significantly lower for larger particles (5.0 μm); the reduction of contamination was observed between H-PAC Off and H-PAC on activities.

For activity S the mean value of 3 runs revealed a reduction of particle pollution when the PAC device was on (Table 2 and Figure 7):-76% ± 7.69% reduction of 0.3 μm particle size-82% ± 13.90% reduction of 0.5 μm particle size-83% ± 16.33% reduction of 1.0 μm particle size-81% ± 9.81% reduction of 5.0 μm particle size

The obtained data showed that the particle contamination abatement was significantly higher for all particle sizes in activity S compared to activity H, with higher values in some cases of “At Rest” monitoring. The above results were achieved regardless of the initial particle contamination value; in fact in three runs with the PAC off, the mean contamination values were respectively 28 × 10^6^, 16.7 × 10^6^ and 20.8 × 10^6^. However, it has enabled us to achieve considerable abatement rates leading to the mean values described above.

### 3.2. Abatement of Microbiological Air Contamination

In order to verify microbial contamination of the air, monitoring was carried out with PAC on and off, thorough two activities with different risk levels as described and shown below:-Risk activity 1, “Professional Dental Hygiene” identified with letter H-Risk activity 2 “Oral Surgery” procedures identified with letter S

The mean microbiological contamination value for Activity H, in 3 runs with PAC off and on respectively for the left and right plates:-161 CFU/m^3^ for right plates (DX) and 204 CFU/m^3^ for left plates (SX) with PAC off-49 CFU/m^3^ for right plates (DX) and 40 CFU/m^3^ for left plates (SX) with PAC on

The mean abatement value of microbiological air contamination is equal to:-69% for right plates (DX)-80% for left plates (SX)

The microbiological air contamination with the PAC off is practically the same for both H and S activities: thus it seems that the contamination sources in both activities might be overlapping. The lower percentage of abatement level in activity S can be attributed to the greater number of subjects present in the room during dental operation. It may partially be hindered by the removal of microbiological air contamination. It is important to mention that the presence of an HEPA 14 filter contributes to a reduction in the density of aerosol particles, which results in a reduction in microbiological contamination.

### 3.3. Sound Results

#### 3.3.1. Sound Pressure Level “At Rest”

During the “At Rest” phase, the sound pressure level was detected at the same points used for detection of particle air contamination, with PAC off and on. Based on attached phonometric data, the spatial variability for each of the three acoustic parameters (Lpk, LE, LFp) with both PAC on and off were measured, as the difference between the mean value of the 6 points and the values obtained for each individual point (dB). The results showed that the noise pollution level was not significantly different regardless of the staff position in the room or whether the PAC was on or off.

The following results were obtained in the “At Rest” phase with PAC off and on:-There is an increase in the mean value of Lpk peak by 3 dBC (4%), while this value was not affected in “At Rest” phase. This increase is due to the presence of operators’ noises during their activities.-There is an increase in the mean value of LE by 9 dBC (13%), which means that the human ear can perceive 13% more noise signal while the PAC was on.-There is an increase in the mean value of LFp magnitude by 8.5 dBA (16%) in the noise perceived by the human ear, which is induced by the PAC device.

The above findings demonstrate that when the only variable was switching the PAC device on or off, there is a significant difference in all three parameters “At Rest” conditions (Lpk, LE, LFp), as was expected.

#### 3.3.2. Sound Pressure Level during “Operational” Phase

Throughout the “Operational” phase, sound pressure level was detected at the same points used for detection of particle air contamination, with the PAC off and on. The following results were obtained for the first monitored dental activity “professional Hygiene” (H).

-The noise, measured as the mean value Lpk peak (dB), was not significantly different in two situations (PAC off and on): due to the fact that, during dental activities, the peak noise of the instruments overlap with that of the PAC with similar peak values, it cancels out the noise effects induced by the PAC device alone.-Even for LE, there was no significant difference in mean values (84.5 dBC with PAC off versus 82.6 dBC with PAC on). This means that the average detected signal energy overlapped. Moreover, in this case, there was a minimal difference due to the overlapping of noises produced by dental instruments with those of the PAC system.-The similarity in noise values is even more marked when LFp was considered. In this case, the mean values were found to be different by 1.2 dBA between the PAC off and on situations. The dental equipment’s noises masked the possible increase in noise induced by the PAC device.

Regarding surgical activity (S), there was a slightly more marked difference in the mean value of all three parameters with the PAC off and on:-3 dBC difference in the mean value of Lpk peak;-4 dBA difference in the mean value of the LE;-5 dBA difference in the mean value of LFp.

## 4. Discussion

The usage of an air-filtering device inside the dental clinic during the operational phase improves the air quality based on the achieved results during At Rest, in which it shows a contamination reduction of 85–64%. This reduction is due to the specific characteristics of this filtering device, such as an air quality indicator to monitor the air in domestic environments, HEPA 14 filter with activated carbon and Photocatalysis, UV function, negative ions and environmental purification up to 1300 cubic meters per hour.

This study revealed the remarkable effect of the PAC device on the reduction of particle contamination by 64–85% At-Rest phase, 49–73% during professional dental hygiene activity and 76–83% during simple surgery procedure.

The effectiveness of the PAC is also confirmed from a microbiological point of view as there is a reduction ranging from 69 to 80% in professional dental hygiene activity and from 62 to 66% during simple surgery activity.

The PAC device does not produce significant additional noise during dental operations, as it is masked by the noise emitted from other devices, despite the fact that an increase from 3 to 9 dB between PAC on and off had been detected At-Rest.

The results obtained are in line with other studies already present in the literature [11,13].

In addition to PPE (Personal Protective Equipment), the use of an airpurifier device with an HEPA 14 filter has been already validated [2,3].

This study was conducted in an operational environment to reflect the real clinical situations during usual daily practice. Thus, this choice made it possible to obtain data in a real situation. Much more reliable results might be achieved if the same study could be conducted in a controlled environment without changing procedure plans during the operational phase.

## 5. Conclusions

In conclusion, this study showed the higher efficacy when PAC was on in pollution abatement, 83% more when the PAC device was off. Additionally, the contamination was reduced by 69–80%. Noise pollution was not noticeable compared to the sounds already present in the clinical environment. The addition of PAC equipment to the already existing safety measures was found to be significantly effective in further microbiological risk reduction.

## Figures and Tables

**Figure 1 ijerph-19-05139-f001:**
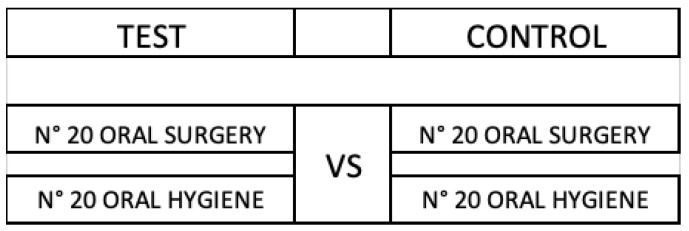
Test groups versus control groups.

**Figure 2 ijerph-19-05139-f002:**
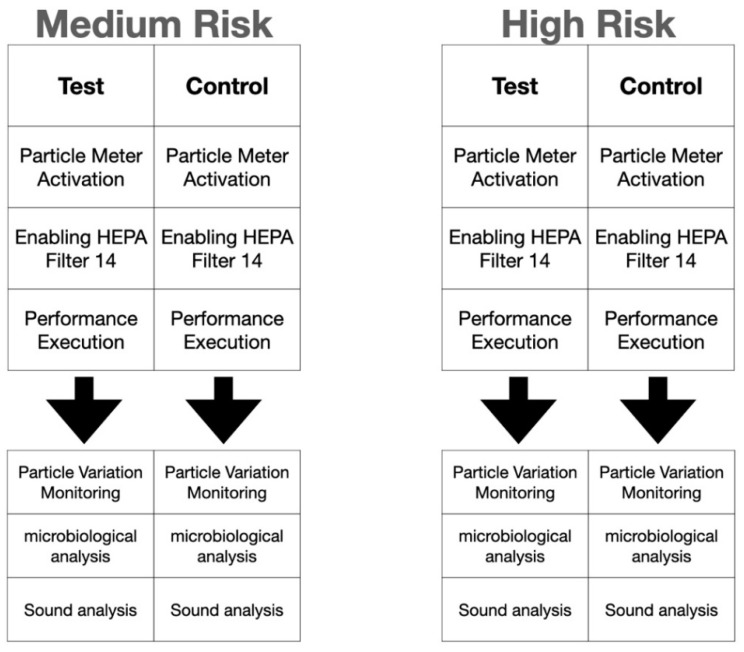
Description of medium- and high-risk groups. HEPA = High Efficiency Particulate Air Filter.

**Figure 3 ijerph-19-05139-f003:**
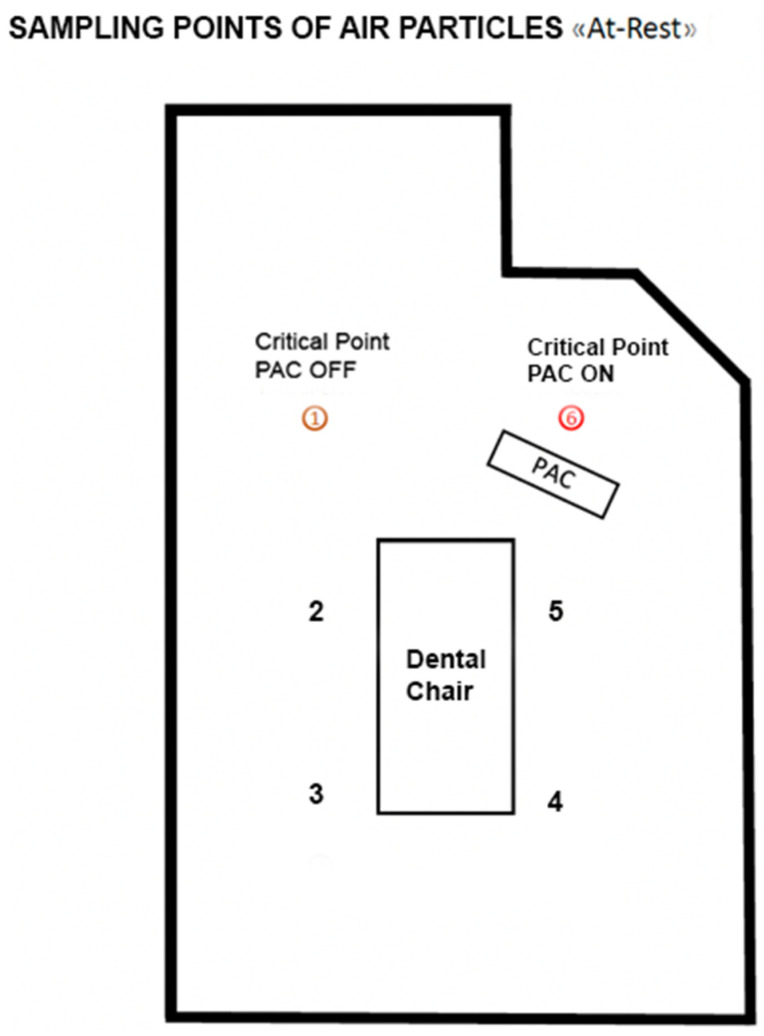
Planimetry of the experiment. PAC = Portable Air Cleaners, the 2 points in red (1 and 6) are the most critical points.

**Figure 4 ijerph-19-05139-f004:**
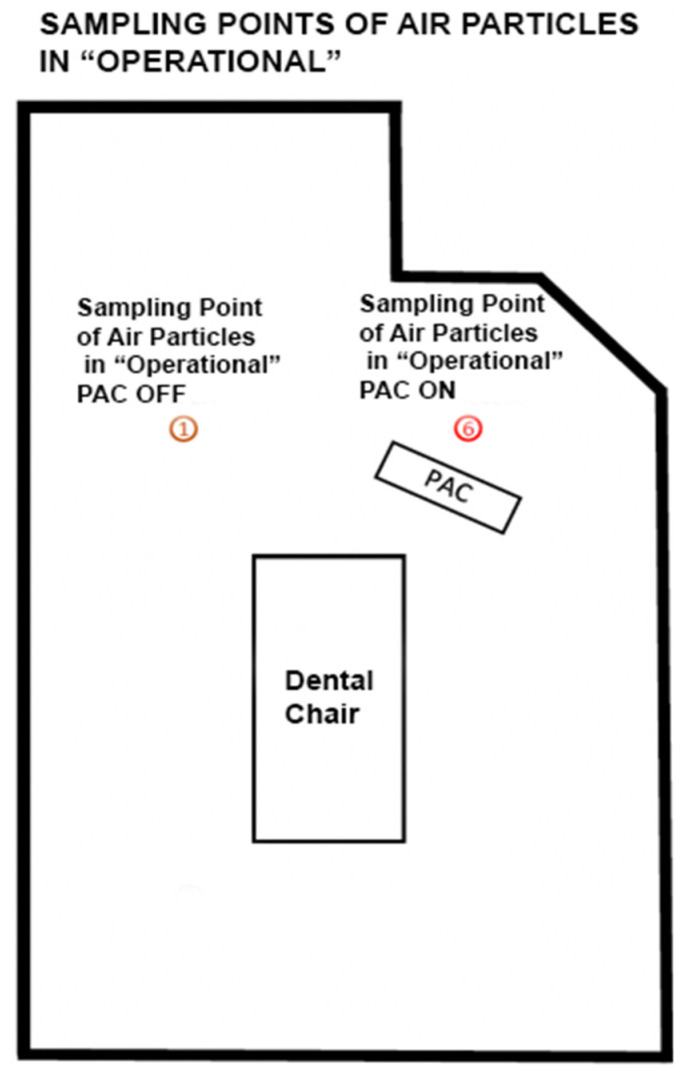
Planimetry of the experiment. PAC = Portable air cleaners.

**Figure 5 ijerph-19-05139-f005:**
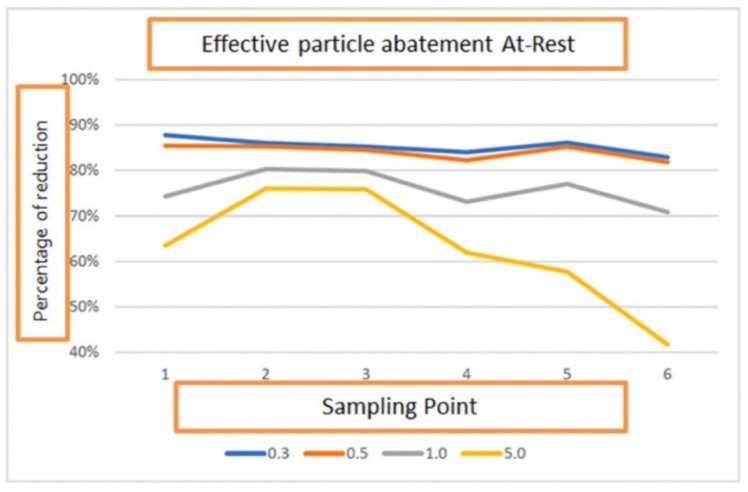
Effective particle abatement At-Rest.

**Figure 6 ijerph-19-05139-f006:**
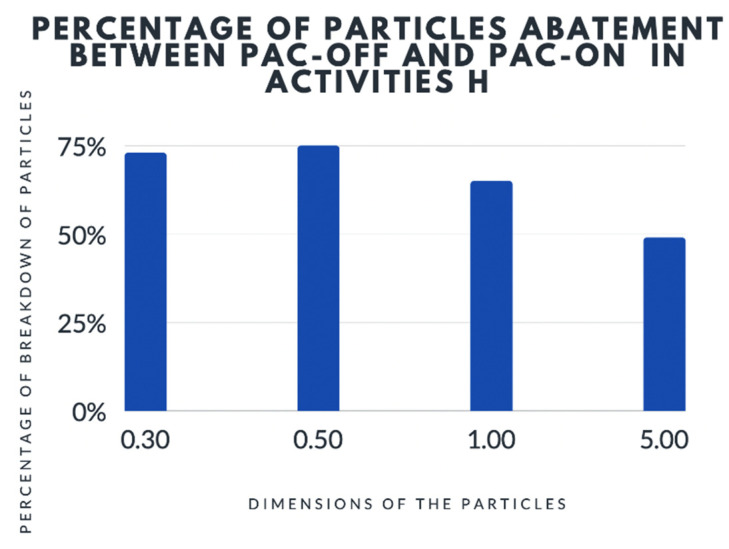
Analysis of particles during activity H with PAC off and on. PAC = Portable Air Cleaners.

**Figure 7 ijerph-19-05139-f007:**
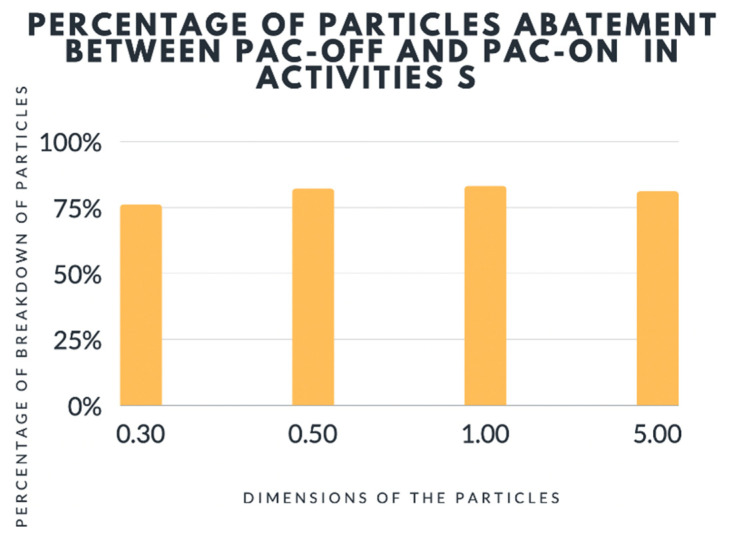
Analysis of particles during activity S with PAC off and on. PAC = Portables Air Cleaners.

**Table 1 ijerph-19-05139-t001:** Analysis of particles during activity H with PAC off and on.

	Percentage of Breakdown			
	Dimension of the Particles (µm)			
Avarage	0.30	0.50	1.00	5.00
Value	73%	75%	65%	49%
	**Percentage of Residual Particles**			
	**Dimension of the Particles (µm)**			
Avarage	0.30	0.50	1.00	5.00
Value	27%	25%	35%	51%

**Table 2 ijerph-19-05139-t002:** Analysis of particles during activity S with PAC off and on.

	Percentage of Breakdown			
	Dimension of the Particles (µm)			
Avarage	0.30	0.50	1.00	5.00
Value	76%	82%	83%	81%
	**Percentage of Residual Particles**			
	**Dimension of the Particles (µm)**			
Avarage	0.30	0.50	1.00	5.00
Value	24%	18%	17%	19%

## Data Availability

The data presented in this study are available on request from the corresponding author.

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
