# Peer review of "The Usage of an Air Purifier Device with HEPA 14 Filter during Dental Procedures in COVID-19 Pandemic: A Randomized Clinical Trial"

_ijerph, 2022, doi:10.3390/ijerph19095139_

Round 1

Reviewer 1 Report

Rating the Manuscript

  • Originality/Novelty: The question is original and well defined but the topic is not adequately explained with regard to COVID. The application of the filter is explained in general, independent of the Covid virus pandemic.
  • Significance: The results are interpreted appropriately and  they are significant.
  • Quality of Presentation: The article is written in an appropriate way and the the data and analyses are presented appropriately.
  • Scientific Soundness: the analyses are performed with the highest technical standards. The methods, tools, software, and reagents are described with sufficient details to allow another researcher to reproduce the results. I am interested in whether the patient signed an informed consent and whether they have been tested for Covid due to the current situation and the possibility of influencing the results
  • Interest to the Readers: the conclusions are interesting for the readership of the Journal
  • English Level: the English language is appropriate and understandable.

Overall Recommendation

  • Accept after Minor Revisions: The paper is in principle accepted after revision based on the previous comments.

Review Report

  • 11. i 12.  cited references are not written in an appropriate way
  • The manuscript’s results are reproducible based on the details given in the methods section
  • the figures/tables/images/schemes are appropriate
  • Materials and methods should at least state that the patient was informed and tested. It is not clear by which criteria the categorization of risk is determined (high, medium…) and which are ‘simple oral surgery procedures’?
  • I recommend that more information be provided in the introduction and discussion about the use of HEPA filters

Author Response

Review  1 Report

i 12.  cited references are not written in an appropriate way

Thanks to the reviewer the correction was made

The manuscript’s results are reproducible based on the details given in the methods section

the figures/tables/images/schemes are appropriate

Materials and methods should at least state that the patient was informed and tested. It is not clear by which criteria the categorization of risk is determined (high, medium…) and which are ‘simple oral surgery procedures’?

We thank the reviewer for the comment the required sections were added as requested:

“A specific informed consent related to each treatment was obtained from the patients prior to perform any dental procedures.” , “simple oral surgery procedure; without usage of instruments that generates aerosol e.g Turbine, handpiece”

I recommend that more information be provided in the introduction and discussion about the use of HEPA filters

We added a new part explaining the filtering device: “The term HEPA stands for high-efficiency particulate air, indicates a particular highly efficient filtration system for fluids, liquids or gases.

The HEPA filter is made up of microfibre filter sheets, thousands of glass fibers that intertwine in multiple layers, separated by aluminum septa. These layers of filter sheets have the task of blocking the polluting particles present in the area to be treated”

Reviewer 2 Report

Please check the  references  and the citations missing (Cherlone et al)  and  not refered in the References list. The same happens to the las  paper # 12 by Ren et al.

Add some  more recently published relevant to the issue,papers

Author Response

All the changes asked by the Reviewer 2 were made in the main text and some of them were also explained below.

This is a paper of 2010. It is impossible to describe protocols  for  Sars CoV 2:  Thank you, we should mention the fact that this article was not used for CoV 2 protocol but as a general reference for microbiological safety measures.

Regarding line 98 : Please consider that the 50 liters/minute mentioned here is regarding the particle counter system ( Lasair III ) which is different from the PAC  device flow rate.

Regarding line 120; is there a possibility to be CFU (colonies forming units)? Thank you for the comment,  Yes CFU stands for Colony Forming Units

Reviewer 3 Report

Understanding the interplay between PAC and dental clinic indoor air quality for protecting public health. Since the air purifier is a device that dilutes indoor air by utilizing airflow, air pollution can spread further if used incorrectly. This paper analyzes how PAC affects dental clinic indoor air quality. However, although this paper deals with aerosol behaviour and air cleaning devices, it lacks an explanation, discussion, and analysis of the experimental design and results about that. In addition, overall, I don’t think this paper is at the level to be published. Even basic things such as specifying the location of the figure, the definition of general abbreviations, and a lot of typos are not prepared. Therefore, I regretfully do not recommend its publication in International Journal of Environmental Research and Public Health.

- Figure 2: What is the difference between Test and Control? And What is the meaning of a bold black box?

- Figure 2 caption: What is the EPA filter 14? According to the definition of the BS EN 1822, High efficiency air filter test standard, class E14 cannot exist. Generally, EPA indicates efficient particulate air filters having classes 12 or less. Classes 13 to 14 should be expressed in HEPA filter. Thus, EPA and HEPA are obviously different.  The definition of “EPA/HEPA = High Efficiency Particulate Air Filter” is also not right.

- Line 91: Please use only one expression Step or Phase. Ex) Step 1 and Step 2.

- Line 97: Please add more detailed information about the particle counter system. If it is a self-manufactured system, authors must show detailed information, and if it is purchased equipment, authors must show an accurate company name and model number.  

- Figure 3: Please unify the font size. And numbers 1 and 6 should be modified so that they can be viewed more clearly.

- Line 120: What is the UFC?

- Line 156: In the evaluation of air cleaning performance, initial aerosol concentration is really important. Authors need to add information on initial aerosol concentrations (PAC off).

Line 161: In the previous study reported that the air cleaning performance of the air purifier changes over time (Int. J. Environ. Res. Public Health 2021, 18(16), 8426). Is the particle removal efficiency constant during the test time? Or is there any change?

- Line 162: What are the critical points? Please define this clearly. Critical points are just only shown in Figure 3, and the authors need to explain why 1 and 6 are critical points (more difficult to clean).

- Line 170: I think it was measured at 6 locations, but where is the information about 6 locations? And How does the concentration of aerosols vary depending on the measurement location?

- Line 178-181, 187-190: Please add to the standard deviation.

- Line 182: Typically, it is much easier to remove large particles. However, in this paper, small particles are more easily removed by PAC. Why is that?

- Line 194: What is the S2, S4, S6? Authors need to define them.

- Results II: Why does PAC reduce the microbial concentrations? In general, air-cleaning devices (air purifiers) are a device with the concept of diluting air by continuously cleaning air, not a device that filters out all pollutants in indoor air. In other words, if the concentration of microorganisms has decreased, it is necessary to confirm that all of the microorganisms are filtered through the filter and whether the CADR of the used PAC is suitable for the space used in the experiment. If not, microorganisms may actually be more easily delivered to the patients due to the flow caused by the air purifiers.

Author Response

Understanding the interplay between PAC and dental clinic indoor air quality for protecting public health. Since the air purifier is a device that dilutes indoor air by utilizing airflow, air pollution can spread further if used incorrectly. This paper analyzes how PAC affects dental clinic indoor air quality. However, although this paper deals with aerosol behaviour and air cleaning devices, it lacks an explanation, discussion, and analysis of the experimental design and results about that. In addition, overall, I don’t think this paper is at the level to be published. Even basic things such as specifying the location of the figure, the definition of general abbreviations, and a lot of typos are not prepared. Therefore, I regretfully do not recommend its publication in International Journal of Environmental Research and Public Health.

- Figure 2: What is the difference between Test and Control? And What is the meaning of a bold black box?

We thank the Reviewer for his comment, all the explanation were added: ““The difference between the test and control group is that in test group PAC device with the Hepa 14 filter were on but for the control group the PAC device with the Hepa 14 filter were off.”

The bold back box was only to emphasize the type of analysis; it is removed in the updated version. 

- Figure 2 caption: What is the EPA filter 14? According to the definition of the BS EN 1822, High efficiency air filter test standard, class E14 cannot exist. Generally, EPA indicates efficient particulate air filters having classes 12 or less. Classes 13 to 14 should be expressed in HEPA filter. Thus, EPA and HEPA are obviously different.  The definition of “EPA/HEPA = High Efficiency Particulate Air Filter” is also not right.

We thank the reviewer for the comment, the correction has been made.

- Line 91: Please use only one expression Step or Phase. Ex) Step 1 and Step 2. We thank the reviewer for the comment, the correction was made

- Line 97: Please add more detailed information about the particle counter system. If it is a self-manufactured system, authors must show detailed information, and if it is purchased equipment, authors must show an accurate company name and model number.  

We thank the reviewer for the comment; the exact type of the device was mentioned in the manuscript.

- Figure 3: Please unify the font size. And numbers 1 and 6 should be modified so that they can be viewed more clearly. Thanks to the reviewer a change was made as requested

- Line 120: What is the UFC? We thank the reviewer for the question, UFC is the Italian version and it stands for Colony Forming Units It is also has been corrected in the main text.

- Line 156: In the evaluation of air cleaning performance, initial aerosol concentration is really important. Authors need to add information on initial aerosol concentrations (PAC off). We thank the reviewer for the comment, we should explain that the initial aerosol contamination was controlled by utilising the PAC device with HEPA 14 filter which was 40 µm/m3.

Line 161: In the previous study reported that the air cleaning performance of the air purifier changes over time (Int. J. Environ. Res. Public Health 2021, 18(16), 8426). Is the particle removal efficiency constant during the test time? Or is there any change?

Thank you for the comment, Yes the particle removal efficiency was constant during the test time and there was no change.

- Line 162: What are the critical points? Please define this clearly. Critical points are just only shown in Figure 3, and the authors need to explain why 1 and 6 are critical points (more difficult to clean). Thanks to the reviewr, the explnation was added: “two critical points (more difficult to clean based on the microbiological contamination results during At REST phase).”

- Line 170: I think it was measured at 6 locations, but where is the information about 6 locations? And How does the concentration of aerosols vary depending on the measurement location? The 6 locations include: point 1 which is the worst of the 6 points when PAC is off (based on the microbiological measurment during AT REST), point 6 which is the worst of the 6 points when PAC is on.

- Line 178-181, 187-190: Please add to the standard deviation.

Thanks for the comment, we added the standard deviation of each section in the Result.

- Line 182: Typically, it is much easier to remove large particles. However, in this paper, small particles are more easily removed by PAC. Why is that? We thank the reviewer, we should mention that our measurements tell about the reduction in number of particles between the room with the pac off and in the room with the pac on. Very probably the larger particles will decrease in a lower percentage for two logical reasons: 1. During the dental procedures there is greater production of smaller particles.  2. that the larger pareticles percipitate in greater concentration even while the PAC device is off, therefore the absolute value of larger particle units reduces and consequently the larger particles are less presented in the environment.

- Line 194: What is the S2, S4, S6? Authors need to define them. We thank the reviewer, the correction was made.

- Results II: Why does PAC reduce the microbial concentrations? In general, air-cleaning devices (air purifiers) are a device with the concept of diluting air by continuously cleaning air, not a device that filters out all pollutants in indoor air. In other words, if the concentration of microorganisms has decreased, it is necessary to confirm that all of the microorganisms are filtered through the filter and whether the CADR of the used PAC is suitable for the space used in the experiment. If not, microorganisms may actually be more easily delivered to the patients due to the flow caused by the air purifiers. Thanks to the reviewr, the explanation was added: “It is important to mention that the presence of HEPA 14 filter contributes to reduce the density of aerosol particles which results in a reduction of microbiological contamination.”

Round 2

Reviewer 3 Report

The authors have carefully revised the paper and I don't have further comments.